# Transgenerational and Molecular Responses to Lanthanum Exposure in a *Spodoptera littoralis*-*Brassica rapa* System

**DOI:** 10.3390/ijms26178462

**Published:** 2025-08-30

**Authors:** Cong van Doan, Sara Bonzano, Massimo E. Maffei

**Affiliations:** Department of Life Sciences and Systems Biology, University of Turin, Via Quarello15/a, 10135 Turin, Italy; sara.bonzano@unito.it

**Keywords:** *Spodoptera littoralis*, *Brassica rapa*, plant–insect interactions, transgenerational effect, calcium signalling, oxidative stress, gene expression

## Abstract

The widespread use of rare earth elements (REEs) in agriculture, particularly Lanthanum (La), raises concerns about their ecological impact on non-target organisms. We investigated the direct and indirect effects of La on the insect pest *Spodoptera littoralis* and its host plant, *Brassica rapa*. Direct exposure to La-supplemented diets reduced larval growth, survival, and egg production. Interestingly, a transgenerational effect was observed, where larvae from La-exposed parents exhibited increased resilience, showing no performance reduction on the same diets. Indirectly, La accumulation in plants mediated a hormetic response in herbivores, increasing larval weight at low concentrations but reducing it at high concentrations, while modulating their oxidative stress and detoxification gene expression. From the plant perspective, La exposure amplified herbivory-induced calcium signalling and altered the expression of key genes related to calcium and reactive oxygen species pathways. These findings reveal the complex ecological risks of La accumulation in agroecosystems, affecting both plants and insects directly and through novel transgenerational effects.

## 1. Introduction

Rare earth elements (REEs), a group of 17 chemically similar metallic elements including Lanthanum (La), are attracting increasing attention due to their crucial role in daily life [1]. In China, for example, farmers use REEs to enhance the yield and biomass accumulation of cereal and vegetable crops [2]. However, the widespread use of REEs has also raised environmental concerns [3]. Their persistence in the environment and potential toxicity to non-target organisms have led to increased ecological concerns [4,5].

Among the REEs, La is one of the most biologically active and abundant. Its effects on plant metabolism are well-documented, including alterations in photosynthetic efficiency, enzyme activity, and redox homeostasis [6,7]. While many studies have focused on the effects of La on plant physiology, its ecological consequences, particularly in multitrophic interactions such as plant–insect interactions or those at higher trophic levels, remain less investigated [8]. Plants exposed to La may experience altered expression of genes involved in oxidative stress response. These changes could influence not only their own growth and defence systems but also affect herbivore performance by altering nutritional quality or chemical defences [9].

*Spodoptera littoralis*, a generalist insect pest, is vulnerable to environmental contaminants transmitted through its diet. Heavy metals and nanomaterials, for example, can disrupt insect development, metabolism, and detoxification pathways [10]. However, few investigations have addressed whether La exerts direct toxicity to insects or if such effects can propagate across generations through physiological preconditioning or epigenetic mechanisms [11]. The consumption of REEs via an artificial diet or plant material containing REEs could disrupt redox homeostasis, impair detoxification pathways, and reduce larval fitness [12]. Moreover, it remains unclear whether REE exposure can modulate insect gene expression related to oxidative stress and detoxification [13]. Insect herbivores feeding on REE-treated plants may be indirectly exposed to these elements via plant tissues, potentially triggering oxidative stress, detoxification responses, and growth impairment [14].

From the plant perspective, we recently demonstrated that important crops like *Brassica rapa* can accumulate La in their tissues upon irrigating plants with increasing concentrations of La [6]. Studies have also begun to explore how REEs interfere with core signalling mechanisms, particularly calcium (Ca^2+^) signalling [15]. Calcium is a central second messenger involved in plant responses to herbivory, wounding, and abiotic stress [16,17]. La application has been shown to impact plant signalling systems, especially calcium signalling, which plays a central role in mediating responses to abiotic stress and herbivory [18,19]. La has the potential to interfere with Ca^2+^ fluxes and alter the spatiotemporal dynamics of signal propagation [15], which could affect how plants perceive and respond to insect attacks.

In this study, we investigated both the direct and indirect effects of La on the growth, physiology, and gene expression of *Spodoptera littoralis* and its host plant, *Brassica rapa*. First, we investigated larval performance, redox physiology, and gene expression in insects feeding on artificial diets supplemented with La. We then examined transgenerational effects by comparing the performance of a second generation of larvae originating from parents that were either exposed or unexposed to La. Finally, we assessed how *Brassica rapa* plants respond to the combined exposure to La and *Spodoptera littoralis* herbivory, focusing on biomass, pigment content, antioxidants, Ca^2+^ signalling, and gene expression. Our findings provide mechanistic insight into the complexity of REEs’ interactions and highlight the potential ecological risks of their accumulation in agricultural systems.

## 2. Results

### 2.1. Lanthanum-Supplemented Artificial Diet (AD) Alters Spodoptera Littoralis Growth, Survival, Metabolism, and Resilience

Larvae fed with artificial diets (ADs) supplemented with increasing levels of La exhibited significantly reduced weight gain (*p* < 0.05) at early time points compared to larvae on the control AD. However, the growth rate itself did not differ significantly among treatments (Figure 1A). After 7 days, a significant reduction in larval weight was observed only at the highest La concentration (10 mM) (Figure 1A). Similarly, a significant reduction in larval survival rate was found only at the 10 mM La concentration across all time points (Figure 1B).

We then assessed the larval physiological responses after 7 days of exposure. Larval protein content was significantly increased in larvae feeding on the AD supplemented with 10 mM of La, while no significant differences were found at the other La concentrations compared to the control (Figure 1C). The peroxide content of larvae did not show any significant differences across all La concentrations compared to the control (Figure 1D).

Based on these findings, we further investigated reproductive parameters in larvae fed with 10 mM of La for 7 days. With respect to controls, the pupation rate significantly increased (Figure 1E), while pupal weight remained unchanged (Figure 1F). Conversely, the proportion of males among emerging adults (Figure 1G) and the total egg production (Figure 1H) both significantly declined.

When second-generation *S. littoralis* larvae from unexposed parents were fed with AD containing 10 mM of La, a significant reduction in larval weight gain was observed. However, this weight reduction was not seen in larvae from parents previously exposed to 10 mM of La when they were also fed AD containing 10 mM of La (Figure 2A). A similar pattern was observed in survival rates (Figure 2B).

### 2.2. Brassica Rapa Grown in the Presence of La Affects Spodoptera Littoralis Metabolism and Development

Feeding experiments on *B. rapa* plants grown in the presence of an increasing La concentration revealed indirect effects on *S. littoralis* performance. Larvae feeding on these plants exhibited hormetic behaviour, showing increased weight gain in plants exposed to 1 µM La of but decreased weight gain when feeding on plants treated with La in 1 mM and 10 mM concentrations, compared to larvae feeding on untreated plants (La0) or the control AD (AD La0) (Figure 3A).

The protein content of larvae feeding on plants was always significantly lower when compared to larvae feeding on an AD (Figure 3B). However, among larvae feeding on plants, La exposure prompted an increase in protein content at all concentrations, particularly at 1 µM of La, when compared to larvae feeding on untreated plants (La0) (Figure 3B).

The total peroxide content in *S. littoralis* larvae increased after feeding on La-exposed plants, especially those treated with 1 µM and 1 mM of La. No significant differences were found in larvae feeding on plants treated with 10 mM of La compared to untreated plants (Figure 3C). Notably, feeding on plants consistently resulted in a stronger peroxide production than feeding on the artificial diet (Figure 3D).

We also examined the gene expression related to oxidative stress in larvae feeding on La-treated plants. Oxidative stress-related gene expression showed a strong upregulation of catalase (*CAT*) compared to larvae fed the artificial diet (Figure 3D). Interestingly, the strongest upregulation of *CAT* was found in larvae feeding on plants exposed to 1 µM of La (Figure 3D). Cu-Zn Superoxide dismutase-like (*SOD*) was slightly upregulated in larvae feeding on untreated plants but was significantly reduced in larvae feeding on La-treated plants (Figure 3D). Finally, Glutathione S-transferase 1-like (*GST1*) gene expression showed a hormetic trend. The gene was upregulated in larvae feeding on untreated plants or those treated with 1 mM of La, but it was downregulated in larvae feeding on plants treated with the other La concentrations, particularly the 1 µM of La treatment (Figure 3D).

### 2.3. La and Herbivory Impact on Brassica Rapa Metabolism

*Brassica rapa* exposure to La for 7 days, combined with 24 h of herbivory by *S. littoralis*, significantly affected plant metabolism. Compared to untreated plants (CTRL), leaf fresh biomass was significantly reduced in plants fed on by *S. littoralis* and treated with 1 mM of La (Figure 4A). Leaf total protein content increased in response to herbivory but was reduced in the 1 µM of La treatment compared to plants fed on by larvae but not treated with La (Figure 4B). A general increasing protein trend was observed with rising La concentration (Figure 4B). Total chlorophyll content (chlorophyll *a* + chlorophyll *b*) increased in *B. rapa* leaves that were both fed on by larvae and exposed to either 1 µM or 10 mM of La. However, no significant differences were found in the 1 mM of La treatment compared to the La0 control (Figure 4C). The carotenoid content of *B. rapa* leaves did not show significant changes in response to either herbivory or La treatment (*p* = 0.475).

### 2.4. La Impacts on B. rapa Calcium Signalling and Gene Expression upon Herbivory

The release of calcium in the cytosol is one of the early events upon herbivory, triggering a cascade of signalling events that lead to plant responses [20]. We evaluated fluorimetric calcium signalling in control *B. rapa* plants (La 0), after mechanical damage (MD), and after feeding by *S. littoralis* in both untreated (HW) and 10 mM of La-treated plants (HW + La) (Figure 5). As expected, mechanical damage (MD) prompted a small increase in fluorescence, which was significantly higher than in the untreated control plants (Figure 5, lower panel). Upon herbivory, a strong cytosolic calcium fluorescence was observed in the proximity of the herbivore wound (Figure 5, HW). This calcium signal was further increased upon herbivory in plants treated with 10 mM of La (Figure 5, HW + La and lower panel).

As a major second messenger, Ca^2+^ mediates a variety of signal transduction pathways that ultimately trigger the expression of genes involved in plant responses to both abiotic and biotic stress [20]. Ca^2+^ sensors or Ca^2+^-binding proteins specifically perceive Ca^2+^ signals and translate them to downstream targets. Therefore, we analyzed the expression of some *B. rapa* genes coding for Ca^2+^-binding proteins including calmodulin (*CaM1*), three calmodulin-like proteins (*CML30*, *CML42* and *CML42*), calcineurin B-like protein 2 (*CBL2*), calcineurin B-like interacting protein kinase 1 (*CIPK1*), and a calcium-dependent protein kinase (*CDPK20*).

Figure 6 shows the heatmap of *B. rapa* leaf gene expression upon herbivory (Figure 6A) and a Euclidean clustering analysis (Figure 6B) as compared to untreated plants. In untreated plants (La 0), herbivory upregulated *B. rapa CBL2* and, to a lesser extent, *CPK20*, *CML42*, *CML30*, and *CIPK1*. Conversely, herbivory downregulated *CML43* and, to a lesser extent, *Cam1* and *CAMK*.

In plants treated with La, herbivory progressively reduced the upregulation of *CBL2* and *CML42*, while the expression of *CPK20* did not change with increasing La concentration (Figure 6A). *CML30* was downregulated in plants treated with La upon herbivory, particularly at 1 µM of La and 10 mM of La concentrations. The expression of *CML43*, *CaM1*, and *CAMK* remained downregulated in plants fed on by larvae and exposed to La, particularly at the 1 mM concentration (Figure 6A).

Cluster analysis of calcium signalling gene expression clearly separated the contribution of *CBL2* from all other genes (Figure 6B). Three subclusters indicate a close Euclidean distance between *CPK20*, *CIPK1* and *CML42* (Figure 6B, 1), *CAMK* and *CaM1* (Figure 6B, 2), and *CML30* and *CML43* (Figure 6B, 3). Appendix A provides the Log2 fold change values and statistical analysis.

### 2.5. La Exposure and Spodoptera Littoralis Herbivory Modulate Oxidative Stress in Brassica Rapa Leaves

Calcium triggers signalling pathways, including those involved in the production of reactive oxygen species (ROS) [20]. Herbivory induced a significant increase in peroxide production in *B. rapa* leaves compared to untreated plants (Figure 7A). However, exposure of plants to La significantly reduced this peroxide accumulation in response to herbivory, with the lowest values found in plants exposed to 1 mM of La (Figure 7A).

We then evaluated the expression of several genes involved in ROS scavenging and production. Glutathione peroxidase 6 (*GPX6*), that represents the plant’s strategy of utilizing the glutathione (GSH) pool to mitigate La-induced oxidative stress, was strongly downregulated in herbivore-fed *B. rapa* leaves, regardless of the plant’s exposure to La (Figure 7B). On the other hand, glutathione reductase 1 (*GR1*) and cytosolic L-ascorbate peroxidase 1 (*APX1*) were downregulated by *S. littoralis* feeding on plants exposed to La, particularly at 1 mM and 10 mM concentrations (Figure 7B). No significant changes were found for the expression of Cu-Zn superoxide dismutase (*SOD*, primers used in our study were designed to detect both the cytosolic and plastidic forms), peroxidase 32 (*POX32*), catalase 2 and 3 (peroxisomal *CAT2*, *CAT3*), or glutathione S-transferase 2 (*DHAR2*).

A Euclidean cluster analysis of *B. rapa* leaf gene expression confirmed the clear separation of the *GPX6* downregulation from all other genes and highlighted the similar regulation of *APX1* and *GR1*. The cluster analysis also showed a close Euclidean distance between *POX32* and *SOD*, and between *CAT3* and *DHAR2*.

## 3. Discussion

This study provides novel and multifaceted insights into the direct and plant-mediated effects of Lanthanum (La) on the generalist herbivore *Spodoptera littoralis* and its host plant, *Brassica rapa*. Our findings demonstrate that La’s effects are not limited to a simple dose–response curve but are context-dependent and multifaceted. La exposure disrupts insect performance, alters physiological and biochemical traits, modulates the expression of key genes in both the insect and the host plant, and triggers distinct calcium signalling responses in *Brassica rapa*. These complex, dose-dependent effects underscore the ecological complexity of rare earth element (REE) interactions in agroecosystems.

### 3.1. La Induces Direct Effects on Spodoptera Littoralis

Our results show that La exerts a direct toxicity at high concentrations, evidenced by a significant reduction in larval growth, survival, and egg production when *S. littoralis* was fed a La-supplemented artificial diet. This finding is consistent with a growing body of the literature on REE toxicity in insect systems. For instance, dietary or topical exposure to REE salts has been shown to cause a significant decrease in larval weight, prolonged development, and reduced pupal emergence in various insect species [21]. In the generalist grasshopper *Melanoplus sanguinipes*, the accumulation of REEs after just four days of feeding resulted in the paralysis of appendages [18]. Furthermore, our observation of impaired reproduction aligns with research on fruit flies (*Drosophila melanogaster*) [19] and silkworms (*Bombyx mori*) [22], which demonstrated a decline in fecundity and egg hatching rates following REE exposure. Larvae fed an AD supplemented with La suffered from REE toxicity; however, the reduced weight gain in larvae fed with La-supplemented artificial diets is likely due to either the direct toxicity of La, changes in the diet’s physical properties, affecting feeding behaviour, or a combination of both factors. Interestingly, when larvae were fed on La-treated *B. rapa* plants, the observed changes were more complex, suggesting a combined effect of La toxicity and plant-mediated defence responses. The observed increase in larval protein content at high La concentrations suggests a physiological stress response. It is plausible that the larvae redirected metabolic resources from growth and development towards the synthesis of detoxification or stress-related proteins, rather than simply benefiting from a nutritional supplement. This is further supported by the plant data, where La treatment, particularly at lower concentrations, led to increased chlorophyll content in La-treated plants, even under herbivory. Consequently, the observed changes in larval physiology on these plants might not be a direct response to a La-supplemented diet, but rather a result of the insect’s interaction with a chemically altered host plant.

### 3.2. Transgenerational Resilience to La

One of the most significant findings of this study is the transgenerational effect of La exposure. We found that larvae from La-exposed parents showed no negative effects when re-exposed to the same high-concentration diet, a stark contrast to the performance of larvae from unexposed parents. This result indicates that the parental exposure to La induced a form of transgenerational plasticity. The precise mechanism for this is not yet known, but it is likely mediated by maternal effects, where the parental generation transmits a physiological or biochemical “preconditioning” to their offspring [20]. It is plausible that this involves the transfer of detoxification-related compounds or the induction of heritable epigenetic changes that confer resilience without altering the underlying genetic code. Our current data are consistent with the hypothesis that the offspring are better equipped to handle the toxic effects of La from the start. This phenomenon represents a novel finding for La, highlighting how environmental contaminants can have long-lasting effects that are not immediately obvious and can even lead to increased resilience in subsequent generations. Further research will elucidate the specific mechanisms, such as analyzing the offspring’s detoxification enzyme activity and investigating epigenetic modifications.

### 3.3. The Indirect (Plant-Mediated) Effects on S. littoralis

Our results show that larvae feeding on La-treated *B. rapa* plants exhibited a hormetic effect, growing better at a low La concentration (1 µM) but being negatively impacted at higher concentrations. This complex response highlights the critical role of the plant as an intermediary between the contaminant and the herbivore. The hormetic effect might be a direct consequence of the altered nutritional quality of the leaves, as evidenced by changes in leaf protein and chlorophyll content. It is possible that low-level stress from La exposure could have triggered a mild defence response in the plant that paradoxically provided a nutritional benefit to the larvae, while high-level stress compromised the plant’s health, making it a poorer food source. Furthermore, we found that La exposure in the plants led to changes in larval oxidative stress gene expression (e.g., *CAT*, *SOD*, *GST1*), providing a clear mechanistic link for the hormetic response. Like heavy metals, REEs can disrupt cellular homeostasis, leading to the overproduction of ROS [22]. Our findings suggest that at low concentrations, the larvae’s own antioxidant systems were able to effectively manage this stress, potentially even leading to a beneficial over-compensation, whereas at high concentrations, the stress overwhelmed the system, resulting in detrimental effects.

### 3.4. La’s Impact on Brassica rapa Metabolism and Defence

Focusing on the plant’s side of the interaction, we showed that La treatment, especially in combination with herbivory, altered key plant parameters like biomass, chlorophyll content, and total protein. Building on our previous findings that *B. rapa* accumulates La, which impairs photosynthesis and nutrient uptake [6], the current study provides critical insight into the underlying defence mechanisms.

The most striking finding is that La interferes with Ca^2+^, a central secondary messenger in plant defence [16]. The altered expression of Ca^2+^-related genes (e.g., *CBL2*, *CMLs*) and the amplified calcium signal upon herbivory strongly support the hypothesis that La dysregulates the plant’s defence system. Mechanistically, this is likely due to the chemical similarity between La^3+^ and Ca^2+^. Lanthanum ions can potentially directly compete with Ca^2+^ for binding sites on crucial calcium-binding proteins like calmodulin (CaM), thereby disrupting downstream signal transduction pathways [15,23]. This mimicry can either block normal calcium-dependent responses or lead to unregulated signalling, essentially “short-circuiting” the plant’s internal communication network.

Interestingly, while herbivory typically induces peroxide production, La exposure significantly reduced this response. This suggests that by compromising calcium signalling and the subsequent production of key defence compounds like peroxides, La may weaken the plant’s ability to mount a full defence. This compromised defence could be one of the reasons for the hormetic effect observed in the herbivores at low concentrations, where the plant’s reduced ability to defend itself inadvertently benefits the insect.

## 4. Materials and Methods

### 4.1. Plant and Animal Material

*Spodoptera littoralis* eggs were kindly provided by the Department of Agronomy, University of Napoli “Federico II”, Italy. Larvae of *Spodoptera littoralis* were reared on an artificial diet (AD) as described by Agliassa et al. [24] with modifications as detailed below. *Brassica rapa* seeds were provided by Deine Gartenwelt, Emsbüren, Germany (https://www.deinegartenwelt.de/, accessed on 28 August 2025). Plants were grown and treated as described below. A Lanthanum (La) suspension was prepared in Millipore water, and the pH of each suspension was adjusted to 5.6 [25].

### 4.2. Artificial Diet and Larval Rearing

To preliminarily assess the effect of La on *S. littoralis* larvae, a range of La concentrations were tested: 0; 0.1; 0.5; 1; 5; 10; 50; 500; and 1000 µM. Based on these trials, we selected three La concentrations for further experiments: 1 µM, 1 mM, and 10 mM. The 1 µM concentration was chosen as a low, environmentally relevant concentration to represent a plausible exposure scenario. The 1 mM concentration was selected because it induced a moderate, yet statistically significant effect on the plants or larvae, serving as a key point in the dose–response relationship. Finally, 10 mM was used as a high concentration to ensure a clear and maximal biological effect, allowing for a comprehensive study of the stress response mechanisms. This selection of concentrations—low, medium, and high—provides a robust framework to evaluate the full range of La’s effects, from subtle impacts to severe stress, and aligns with dose–response principles used in ecotoxicological studies.

The artificial diet (AD) was composed of 200 g of ground-soaked cannellini beans (Fagioli Cannellini, Centallo, Italy), 2.66 g of ascorbic acid (Sigma-Aldrich, Milan, Italy), 2.66 g of ethyl hydroxybenzoate (Fluka, Milan, Italy), 8 g of plant agar (Duchefa, Haarlem, The Netherland), and 350 mL of distilled water. The pH of the diet was adjusted to 6.0 prior to use to ensure the stability of the vitamin C and other essential nutrients. The pH adjustment was performed immediately before the diet was provided to the larvae. This minimized the exposure time of the vitamin C to the pH 6.0 environment [26]. The AD was supplemented with La at the concentrations listed above. The control was represented by AD without La. The artificial diet was stored at −20 °C until use.

Third-instar *S. littoralis* larvae were reared on AD without La for 7 days before the experiment. Pre-weighed third-instar larvae were then individually placed in square Petri dishes (9 × 9 cm) with 2 g of AD containing the different La concentrations for 7 days. The AD was replaced daily to ensure a fresh and sufficient food source for the larvae. For each treatment, we performed at least four biological replicates. Larval survival and growth rate were monitored after 48 h, 72 h, and 7 days. At the end of the experiment, larvae were collected, immersed in liquid nitrogen, and stored for further analysis

### 4.3. Transgenerational Effects of Lanthanum on Spodoptera littoralis

To investigate transgenerational effects, we offered *S. littoralis* larvae an artificial diet (AD) containing either 0 or 10 mM of La until pupation. We then evaluated the number of pupae, the number of emerging adults, the male-to-total-adult ratio, and the number of eggs deposited by the moths.

The second generation, which originated from parents exposed to either the La-containing or control AD, was subsequently fed on a diet with either 0 or 10 mM of La. We assessed the survival rate and growth rate after 48 h.

### 4.4. Brassica rapa–Spodoptera Littoralis Interactions in the Presence of La

Twenty-one-day-old *Brassica rapa* plants were irrigated with La at concentrations of 0, 1 µM, 1 mM, and 10 mM for 7 days as previously described [6]. Briefly, we used 21-day-old *B. rapa* plants in a pot experiment. The study utilized a completely randomized design with the three different concentrations of a La suspension. Each concentration group, as well as a control group, included six biological replicates, with a single plant per pot. Control plants were watered with distilled water instead of the La suspension. The plants were grown in Traysubstrat Soil (Klasmann-Deilmann GmbH, Geeste, Germany). This substrate is a blend of various components, including sphagnum peat, wood fibre, perlite, clay, and coir, and contains a wetting agent to ensure uniform water absorption. The soil itself was confirmed to be free of lanthanum. Third instar pre-weighed *S. littoralis* larvae were then allowed to feed on the treated plants for 24 h. The experiment followed a completely randomized design with five to six biological replicates. Each biological replicate consisted of a single plant per pot. At the conclusion of the experiment, larvae and leaves were then weighed, collected, frozen in liquid nitrogen, and stored at −80 °C for further analysis.

### 4.5. Determination of Total Protein Content

Total protein from *B. rapa* leaves and from whole *S. littoralis* larvae was measured by using the Pierce™ Bradford Protein Assay Kit (Thermo Scientific, Waltham, MA, USA) according to the manufacturer’s instructions. Absorbance readings were taken at 595 nm using a microplate reader. A standard curve was generated using bovine serum albumin (BSA) to accurately quantify the protein concentration in the samples.

### 4.6. Determination of Chlorophyll and Carotenoid Content

Total chlorophylls and carotenoids were extracted and quantified as previously described [6], with the following modifications: 100 mg of ground fresh leaves were extracted with 1mL of 95% ethanol. The Chl *a*, Chl *b*, and Car contents were measured using a UV1280 spectrophotometer (Shimadzu, Kyoto, Japan) at 664 nm, 649 nm, and 470 nm, respectively. All pigment concentrations were expressed on a fresh weight (FW) basis. At least three biological replicates were used.

### 4.7. Determination of Total Peroxide Content

The total peroxide content was measured using a PEROXsay Assay Kit (Sigma-Aldrich, St. Louis, MO, USA) as previously described [6]. Briefly, 100 mg of freshly ground leaves and larvae from at least three biological replicates were extracted with 1 mL of Milli-Q water. Absorbance was read at 600 nm.

### 4.8. Determination of Intracellular Calcium Variations

For the Ca^2+^ imaging, Calcium Orange dye (stock solution in DMSO, Molecular Probes, Leiden, The Netherlands) was diluted to a final concentration of 5 µM in 5 mM MES-Na buffer (pH 6.0) and applied to attached leaves as described in detail in [27]. Fluorescence intensity in *B. rapa* leaves of plants grown with or without 10 mM of La was measured using a Leica TCS SP5 (Leica Microsystems Srl, Milan, Italy) multiband confocal laser scanning microscope (CLSM), following the method described previously [27]. For image analysis, the parameters were set to 1024 × 1024 pixels, 8 bits per sample, and 1 sample per pixel. At least three images were captured per leaf, with a minimum of three biological replicates were analyzed for each treatment. The images, generated by the FluoView software (version 3.2.1), were analyzed using NIH Image software (version 1.63), as previously demonstrated [28]. For quantification, the mean fluorescence intensity was measured for a minimum of at least 5 regions of interest (ROIs) per image, and the data were then normalized to the control group to determine the relative change in calcium levels.

### 4.9. RNA Extraction, cDNA Synthesis, and qRT-PCR Assays

Total RNA was isolated and purified from 100 mg of ground material from *B. rapa* leaves and *S. littoralis* larvae by using Peqlab PeqGOLD TriFast reagent (VWR Avantor, Radnor, PA, USA). The total RNA extract was quantified by using a BioSpec-nano spectrophotometer (Shimadzu, Kyoto, Japan). For cDNA synthesis, 500 ng of total RNA was retrotranscribed into cDNA using qScript Ultra Supermix (Quantabio, Beverly, MA, USA), according to the manufacturer’s instructions.

Quantitative Real-Time Polymerase Chain Reaction (qRT-PCR) analysis was performed using a QuantStudio 3 Real-Time PCR System (Applied Biosystems, Foster City, CA, USA). The reaction mixture consisted of 5 µL of 2X Perfecta SYBR Green Fastmix qPCR Master Mix (Quantabio, Beverly, MA, USA), 0.25 µL of cDNA, and 0.01 nmol of primers (Integrated DNA Technologies, Coralville, IA, USA). ROX was used as an internal loading standard. The reaction was performed as described by [6]. The expression stability of the reference genes (*GADPH* for *B. rapa* and *ACT* for *S. littoralis*) was validated across all experimental conditions to ensure reliable normalization. Primer efficiency for each target gene was also confirmed to be within the acceptable range of 90–110% through a standard curve analysis, which is essential for accurate quantification using the ΔΔCt method. The relative transcript level of each gene was calculated using the ΔΔCt method. The expression of the *GADPH* gene was used as a reference for *B. rapa* leaves, while the *ACT* gene served as the reference for *S. littoralis* larvae. Primers used for qRT-PCR were designed using Primer3 (https://primer3.ut.ee/, accessed on 28 August 2025) and are listed in Appendix A.

### 4.10. Statistical Analyses

All statistical analyses were performed using SPSS software, version 29. Data are expressed as mean values ± standard error. Differences between the La application and control groups were assessed by a one-way ANOVA followed by Tukey’s post hoc test. A paired *t*-test with Bonferroni-adjusted probability was also used where appropriate. Cluster analysis was performed using Euclidean distances and the average linkage method with Systat^®^ 10. Raw data are reported in Appendix A.

## 5. Conclusions

This study demonstrates that the rare earth element Lanthanum (La) has complex and non-linear effects on a model plant–insect system, extending beyond simple dose-dependent toxicity. Our findings reveal a multifaceted interaction where La directly impairs the performance of the generalist herbivore *Spodoptera littoralis* while also inducing a novel transgenerational resilience that may be a form of physiological preconditioning. Furthermore, La’s effects are significantly modulated by the host plant, *Brassica rapa*, which, at low concentrations, mediates a hormetic response in the herbivore, unexpectedly improving its performance. These plant-mediated effects are linked to La’s ability to compromise plant vitality and disrupt critical calcium and oxidative stress signalling pathways.

The implications of these results for agricultural systems are significant and deserve careful consideration. The increasing use of REEs like La as fertilizers or growth enhancers may have unintended consequences that could complicate integrated pest management strategies. The discovery of transgenerational resilience suggests that initial exposure to La could create a selective pressure for the evolution of resistant insect populations, rendering future pest control efforts less effective. Moreover, the hormetic effect highlights a paradoxical scenario where low-level La contamination could, in fact, lead to an increase in pest performance and crop damage. Therefore, our findings underscore the need for a cautious approach to REE application in agriculture and emphasize the importance of further research to fully understand the long-term ecological risks and complex, multi-trophic interactions in contaminated agroecosystems.

## Figures and Tables

**Figure 1 ijms-26-08462-f001:**
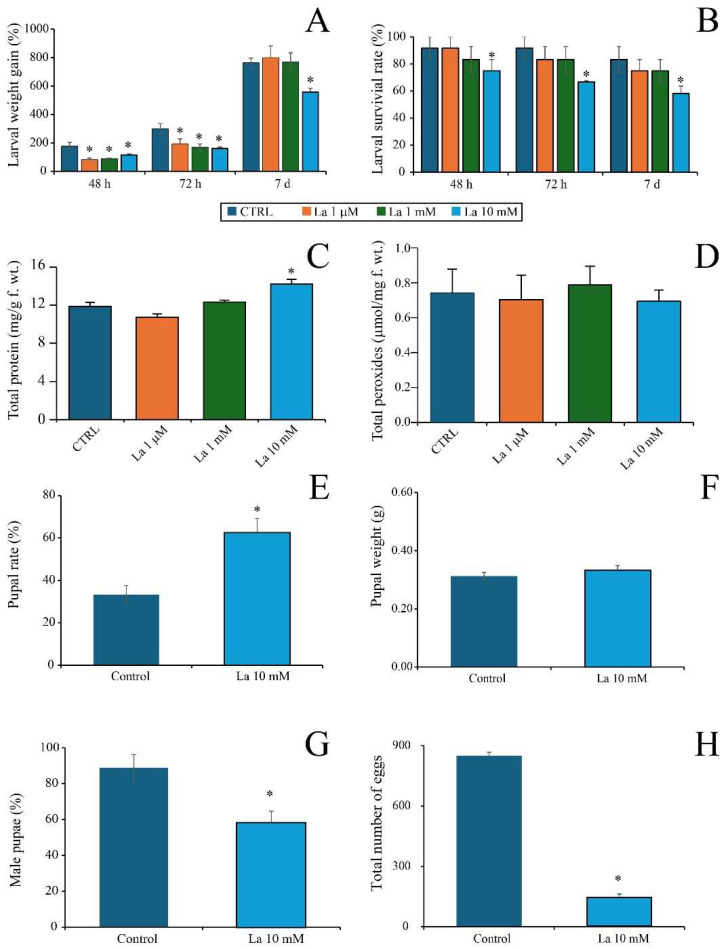
Effect of artificial diet (AD) supplemented with Lanthanum on *Spodoptera littoralis*. (**A**) Time course of larval weight gain and (**B**) larval survival rate. (**C**) Total protein content and (**D**) total peroxide content in larvae after feeding for 7 days on AD supplemented with different La concentrations. (**E**) Pupation rate, (**F**) pupal weight, (**G**) percentage of male pupae, and (**H**) total number of eggs of *S. littoralis* after 7 days of feeding on AD supplemented with 10 mM of La. The metric bars indicate the standard error (*n* = 3–9). An asterisk (*) indicates a significant difference (*p* < 0.05) with respect to the relative control.

**Figure 2 ijms-26-08462-f002:**
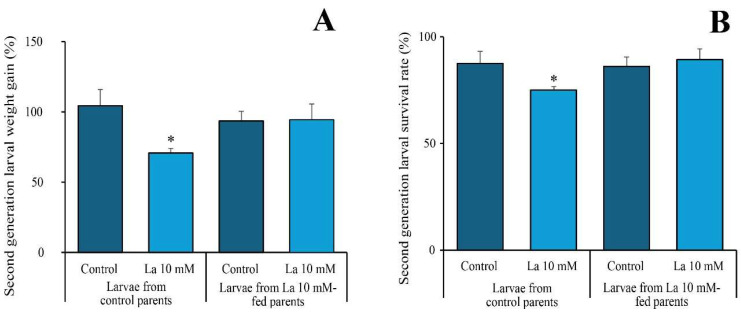
Second-generation *Spodoptera littoralis* larvae from parents fed an AD supplemented with La exhibit increased resistance. (**A**) Larval weight gain and (**B**) larval survival rate in the second generation. Data are mean ± SE (*n* = 6–9). Asterisk (*) indicates significant difference (*p* < 0.05) with respect to the relative control.

**Figure 3 ijms-26-08462-f003:**
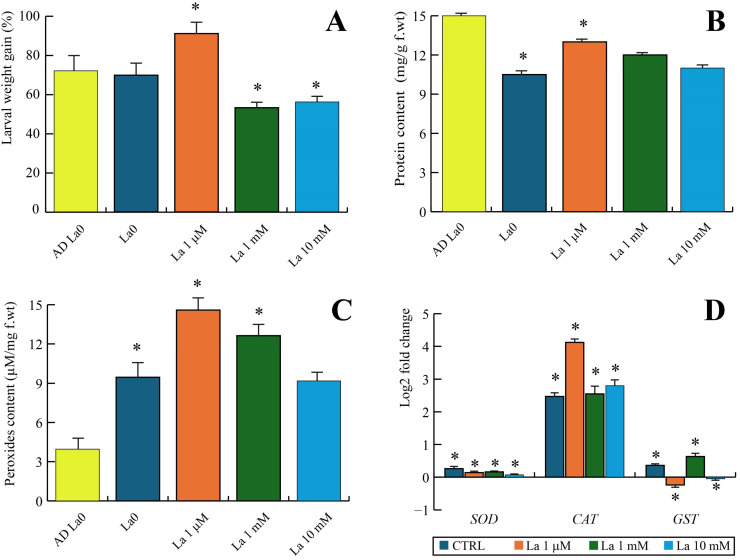
Effects on *Spodoptera littoralis* larvae after feeding on *Brassica rapa* treated with different La concentrations. (**A**) Percentage of weight gain in larvae after a 7-day feeding period. The yellow bar represents the artificial diet control (AD La0). The blue bar represents larvae fed on plants treated with distilled water (La0), which serves as the plant-fed control. The orange, green, and light blue bars represent larvae fed on plants treated with 1 µM, 1 mM, and 10 mM of La, respectively. (**B**) Total protein measured in larvae fed on the different diets, expressed in mg/g f.wt. The bar colours and conditions correspond to those in panel (**A**). (**C**) Total peroxide measured in larvae fed on the different diets, expressed in mg/g f.wt. The bar colours and conditions correspond to those in panel (**A**). (**D**) Log2 fold change in gene expression for three genes, *SOD* (superoxide dismutase), *CAT* (catalase), and *GST* (glutathione S-transferase), in larvae fed on La-treated plants compared to those fed the plant-fed control (La0). The asterisk (*) denotes a significant difference (*p* < 0.05) between a given treatment and its respective control. The error bars indicate the standard error (*n* = 3–5).

**Figure 4 ijms-26-08462-f004:**
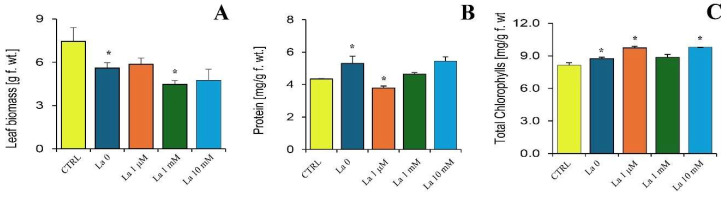
*Brassica rapa* responses to the combined effect of La and herbivory by *Spodoptera littoralis*. (**A**) Leaf fresh biomass, (**B**) total protein content, and (**C**) total chlorophyll content (chlorophyll *a* + chlorophyll *b*). Metric bars indicate standard error (*n* = 3–5). Asterisk (*) denotes a significant difference (*p* < 0.05) with respect to the relative control.

**Figure 5 ijms-26-08462-f005:**
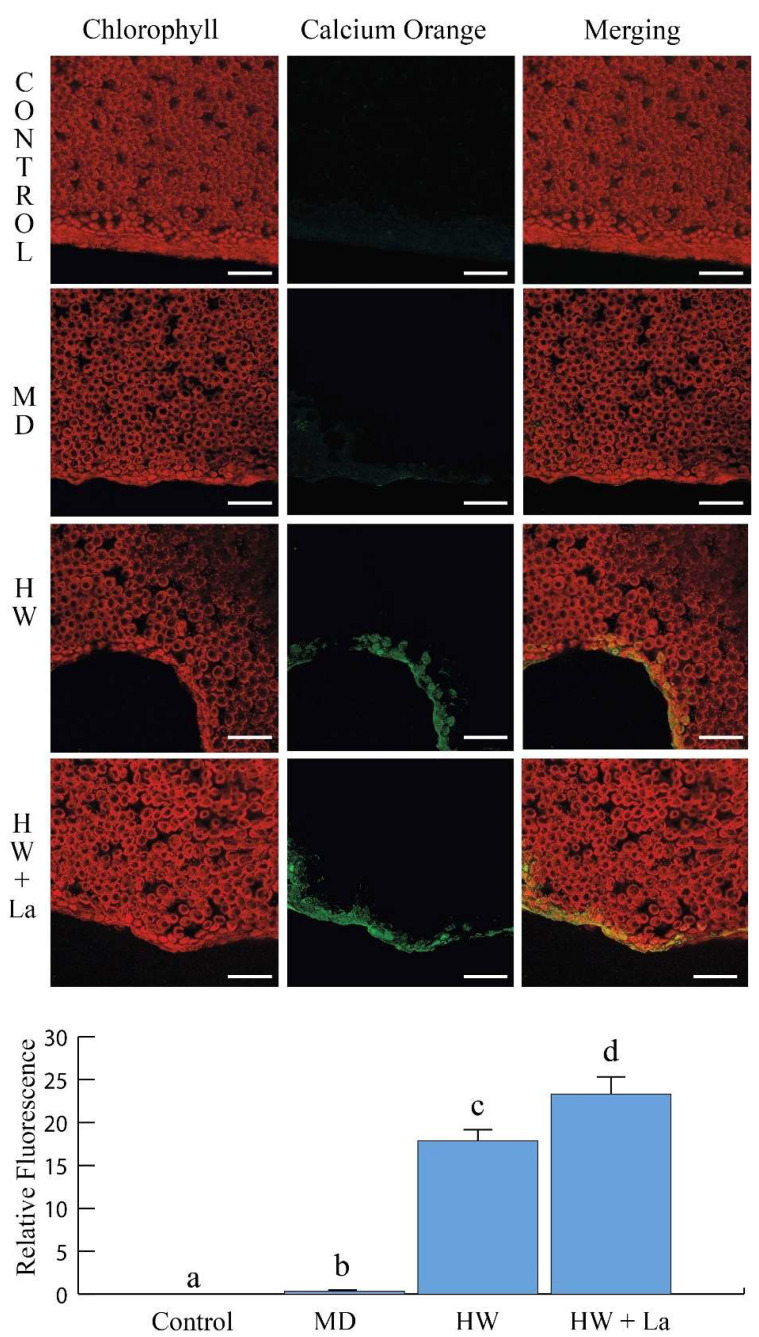
Intracellular Ca^2+^ influx in *Brassica rapa* leaves treated with 10 mM of La upon herbivory. Leaves were subjected to either mechanical damage (MD) or herbivory by *Spodoptera littoralis* in untreated plants (HW) and in plants treated with 10 mM of La (HW + La). False-colour image reconstructions of intracellular Ca^2+^ fluorescence after application of Calcium Orange. Scale bars are 200 µm. The data in the lower panel represent the means and standard errors of the relative quantification levels (*n* = 5). Different letters indicate significant differences (*p* < 0.05).

**Figure 6 ijms-26-08462-f006:**
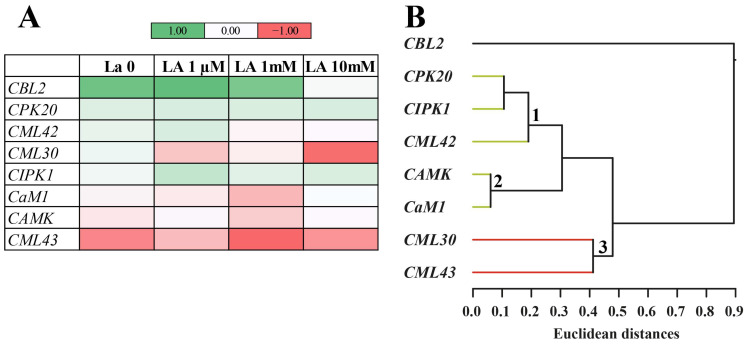
Expression of genes involved in Ca^2+^ signalling in *B. rapa* leaves exposed to La and fed on by *S. littoralis*. (**A**) Heatmap showing the visual representation of gene expression (Log2 fold change) in *B. rapa* leaves from plants fed on by *S. littoralis* compared to untreated plants (La0) and plants exposed to increasing La concentrations. (**B**) Euclidean cluster analysis with the average linkage method. The analysis reveals two major clusters: one represented by *CBL2* and a second comprising the remaining genes. Three subclusters are present within the second cluster (see text for details). The Log2 fold change values and statistical analyses are provided in Appendix A.

**Figure 7 ijms-26-08462-f007:**
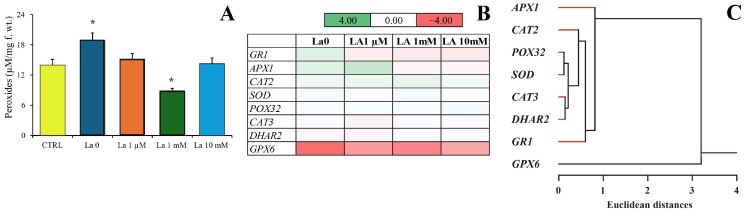
Peroxide production and expression of genes involved in ROS signalling in *B. rapa* leaves exposed to La and fed on by *S. littoralis*. (**A**) Total peroxide content of *B. rapa* leaves from untreated control plants (CTRL), herbivore-fed plants (La0), and La-exposed, herbivore-fed plants. (**B**) Heatmap showing the gene expression (Log2 fold change) in *B. rapa* leaves from plants fed on by *S. littoralis* compared to untreated plants in the absence (La0) and presence of increasing La concentrations. (**C**) Euclidean cluster analysis with the average linkage method shows two major clusters: one represented by *GPX6* expression and a second comprising the remaining genes. In the second cluster, *APX1* and *GR1* gene expression is clustered distinctly from the other genes (see text for details). The Log2 fold change values and statistical analyses are provided in Appendix A. Asterisk (*) denotes a significant difference (*p* < 0.05) with respect to the relative control.

## Data Availability

Data are available on request.

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
