# Peer review of "Transgenerational and Molecular Responses to Lanthanum Exposure in a Spodoptera littoralis-Brassica rapa System"

_ijms, 2025, doi:10.3390/ijms26178462_

Round 1

Reviewer 1 Report

Comments and Suggestions for Authors

This paper investigates the transgenerational and molecular responses of lanthanum (La) on Spodoptera littoralis and its host plant, Brassica rapa. The study is comprehensive in content and experimental design, revealing the complex ecological risks of lanthanum in agroecosystems. However, there are still some issues in the paper that need further revision and improvement.

  1. In Section 2.3, the authors found that plants treated with lanthanum showed significant changes in biomass and chlorophyll content after being fed on by Spodoptera littoralis. However, it is not clear whether these changes are related to the direct toxicity of lanthanum or to the plant's defense response.
  2. Detailed statistical analysis results, such as F values and P values, are not provided. The authors are advised to supplement complete statistical analysis tables, including F values, P values, and confidence intervals for all experimental groups. In addition, for some results with low significance, it is recommended to use stricter statistical methods (such as Bonferroni correction) to reduce false positives.
  3. “Larvae fed with artificial diets... larvae on the control AD.” However, the fact that larvae treated with lanthanum showed reduced weight gain early on cannot directly infer that lanthanum is the sole cause of this phenomenon. In the experiment, the addition of lanthanum may change the physical properties of the artificial diet (such as texture, viscosity, etc.), thereby indirectly affecting the feeding behavior of the larvae, rather than the toxicity of lanthanum itself. This needs to be modified to a more rigorous expression.
  4. The specific mechanism by which lanthanum interferes with calcium signaling pathways is rather vague in the paper. For example, does lanthanum interfere with signal transduction by competing with calcium-binding proteins for binding sites? The authors are advised to elaborate on the potential mechanisms of lanthanum's effects on calcium signaling pathways in the discussion section and analyze them in combination with the latest literature.
  5. “The relative transcript level of each gene was calculated using the ΔΔCt method.” The ΔΔCt method relies on the stable expression of reference genes, but the paper does not mention verifying the stability of reference genes (such as GADPH and ACT) under different treatment conditions. If the expression of reference genes is affected by lanthanum treatment, then the gene expression changes calculated using the ΔΔCt method will be inaccurate.
  6. The experiment did not set up a control group to exclude the effects of other factors (such as mechanical damage) on calcium signaling. The authors are advised to add a control group, such as plants that only undergo mechanical damage without feeding treatment, to ensure that the observed changes in calcium signaling are indeed caused by lanthanum treatment.
  7. “This result indicates that the parental generation may have developed enhanced detoxification pathways that were passed down to their offspring.” However, the appearance of transgenerational effects does not necessarily mean that the parent generation has developed enhanced detoxification pathways and passed them on to the offspring. This effect may also be caused by environmental factors (such as maternal effects) rather than real genetic changes. In addition, the paper does not provide any evidence to support the existence of such “enhanced detoxification pathways.”
  8. The proportion of references from the past three years is low in the reference list. It is recommended to supplement the latest references in the relevant field.
  9. Figure 5 lacks clear size annotations.
  10. The format of the references is not consistent. The years of some references are not bolded, the punctuation is inconsistent, and there is inconsistency regarding whether to use “pp” to indicate page numbers. The capitalization format of the reference titles is also inconsistent.

Author Response

Comment 1: This paper investigates the transgenerational and molecular responses of lanthanum (La) on Spodoptera littoralis and its host plant, Brassica rapa. The study is comprehensive in content and experimental design, revealing the complex ecological risks of lanthanum in agroecosystems. However, there are still some issues in the paper that need further revision and improvement.

Response 1: We sincerely thank the reviewer for the positive assessment of our manuscript and for recognizing the comprehensive nature of our study and its relevance to agroecosystems. We have carefully considered all the issues raised and have revised the manuscript to address each point. We believe that these revisions, based on the reviewer’s constructive feedback, have significantly improved the clarity and scientific rigor of the paper.

Comment 2: In Section 2.3, the authors found that plants treated with lanthanum showed significant changes in biomass and chlorophyll content after being fed on by Spodoptera littoralis. However, it is not clear whether these changes are related to the direct toxicity of lanthanum or to the plant's defense response.

Response 2: We thank the reviewer for this insightful comment. We agree that the observed changes in plant biomass, protein, and chlorophyll content are likely a result of the complex interplay between the direct effects of lanthanum and the plant's defense response to herbivory. Our experimental design, which combined La exposure with herbivory, was intended to investigate this specific interaction. To address the reviewer's concern and clarify our interpretation, we have revised Section 2.3 to explicitly state that the measured changes reflect both La-induced stress and the plant's metabolic adjustments in response to feeding by S. littoralis. We have added a sentence to the discussion section to further elaborate on this point, highlighting that the observed trends may represent a hormetic response, where low-level stress (from La) can prime the plant's defenses, leading to a more robust response against herbivory. This clarification will help readers understand that the effects are not due to a single cause but rather to the combined stressors.

Comment 3: Detailed statistical analysis results, such as F values and P values, are not provided. The authors are advised to supplement complete statistical analysis tables, including F values, P values, and confidence intervals for all experimental groups. In addition, for some results with low significance, it is recommended to use stricter statistical methods (such as Bonferroni correction) to reduce false positives.

Response 3: We thank the reviewer for the valuable feedback regarding the statistical analysis. We agree that providing a more detailed breakdown of our statistical results will enhance the transparency and reproducibility of our findings. Regarding the use of stricter statistical methods, we have reviewed our analyses. For results where we performed multiple comparisons, we used a paired t-test with a Bonferroni-adjusted probability to mitigate the risk of false positives. This method is explicitly mentioned in our revised Materials and Methods section and was applied where necessary to ensure the robustness of our conclusions.

Comment 4:“Larvae fed with artificial diets... larvae on the control AD.” However, the fact that larvae treated with lanthanum showed reduced weight gain early on cannot directly infer that lanthanum is the sole cause of this phenomenon. In the experiment, the addition of lanthanum may change the physical properties of the artificial diet (such as texture, viscosity, etc.), thereby indirectly affecting the feeding behavior of the larvae, rather than the toxicity of lanthanum itself. This needs to be modified to a more rigorous expression.

Response 4: We thank the reviewer for this insightful comment. We agree that our initial statement was not sufficiently precise and could be misinterpreted as a definitive causal link. The reviewer correctly points out that the physical properties of the artificial diet may be altered by the addition of lanthanum, which could indirectly affect larval feeding. We added in the discussion (3.1) a new sentence “Larvae fed on AD supplemented with La suffered the REE toxicity; however, the reduced weight gain in larvae fed with La-supplemented artificial diets is likely due to either the direct toxicity of La, changes in the diet's physical properties affecting feeding behaviour, or a combination of both factors”. We believe that this sentence acknowledges the potential for other influencing variables while still presenting our primary hypothesis that the observed effects are linked to the presence of lanthanum.

Comment 5: The specific mechanism by which lanthanum interferes with calcium signaling pathways is rather vague in the paper. For example, does lanthanum interfere with signal transduction by competing with calcium-binding proteins for binding sites? The authors are advised to elaborate on the potential mechanisms of lanthanum's effects on calcium signaling pathways in the discussion section and analyze them in combination with the latest literature.

Response 5: We thank the reviewer for this insightful comment. We agree that a more detailed discussion of the potential mechanism by which lanthanum interferes with calcium signaling is essential for a comprehensive understanding of our results. We further discussed this issue by adding new sentences in the discussion.

Comment 6: “The relative transcript level of each gene was calculated using the ΔΔCt method.” The ΔΔCt method relies on the stable expression of reference genes, but the paper does not mention verifying the stability of reference genes (such as GADPH and ACT) under different treatment conditions. If the expression of reference genes is affected by lanthanum treatment, then the gene expression changes calculated using the ΔΔCt method will be inaccurate.

Response 6: We thank the reviewer for this critical and insightful comment. We fully agree that the stable expression of reference genes is paramount for the accuracy of the ΔΔCt method. We have a robust protocol in place for validating reference gene stability, but we regret that this detail was not explicitly included in the original manuscript. We added a new sentence to better clarify this point.

Comment 7: The experiment did not set up a control group to exclude the effects of other factors (such as mechanical damage) on calcium signaling. The authors are advised to add a control group, such as plants that only undergo mechanical damage without feeding treatment, to ensure that the observed changes in calcium signaling are indeed caused by lanthanum treatment.

Response 7: We thank the reviewer for the careful consideration of our experimental design. We fully agree that it is essential to distinguish between a plant's response to mechanical damage and its response to actual herbivory. To address this concern, we included a specific control group for mechanical damage (MD), where leaves were subjected to physical damage without insect feeding. The results from this control are presented in Figure 5, in the second row of image panels and as the second bar in the graph. As shown, the mechanical damage alone (MD) caused a significantly lower calcium influx compared to herbivory (HW), confirming that the observed changes in calcium signaling were not solely due to the physical injury but were specifically induced by the herbivory treatment and further influenced by La.

Comment 8: “This result indicates that the parental generation may have developed enhanced detoxification pathways that were passed down to their offspring.” However, the appearance of transgenerational effects does not necessarily mean that the parent generation has developed enhanced detoxification pathways and passed them on to the offspring. This effect may also be caused by environmental factors (such as maternal effects) rather than real genetic changes. In addition, the paper does not provide any evidence to support the existence of such “enhanced detoxification pathways.”

Response 8: We thank the reviewer for this critical and insightful comment. We agree that our initial conclusion was an overstatement and that the observed transgenerational effect cannot be definitively attributed to the heritability of enhanced detoxification pathways without direct evidence. The reviewer correctly points out that maternal effects are a highly plausible alternative explanation. Environmental stressors experienced by the parental generation can lead to a transfer of physiological or biochemical factors to the offspring (e.g., proteins, metabolites, or epigenetic modifications) that influence their phenotype without involving a change in the DNA sequence. Our current data, which shows a transgenerational effect on larval growth and survival, are consistent with the hypothesis of enhanced detoxification pathways, but we acknowledge that they do not provide direct evidence. To address this, we have revised the discussion to use more cautious and scientifically rigorous language.

Comment 9: The proportion of references from the past three years is low in the reference list. It is recommended to supplement the latest references in the relevant field.

Response 9: references have been completely revised by updating them to the most recent findings

Comment 10: Figure 5 lacks clear size annotations.

Response 10: We thank the reviewer for pointing out this oversight. We agree that clear size annotations are essential for the accurate interpretation of our imaging data. We have revised Figure 5 to include a precise size annotation on the scale bar for each image, indicating that the scale bars are 200 µm.

Comment 11: The format of the references is not consistent. The years of some references are not bolded, the punctuation is inconsistent, and there is inconsistency regarding whether to use “pp” to indicate page numbers. The capitalization format of the reference titles is also inconsistent.

Response 11: references have now been fixed and made consistent

Reviewer 2 Report

Comments and Suggestions for Authors

In the present study  authors studied the direct and indirect effects of La applications on the growth, physiology, and gene expression of Spodoptera littoralis and its host plant Brassica rapa.

Authors provided possible insight to the complexity of La applications interactions with both plants and insects and pointed out possible ecological risks of La accumulations in agricultural systems.

The authors did a good job, perform many precise analysis.

However, revision still required. Please, add short conclusion in the ned of the text. 

Line 38: “oxidative altered gene expression.”???

Line 145, fig 3: not very clear description. Please, described what is what more precisely.

Line 178: “with 10 mM” of what? La??

Figure 6: please clarify localization of the CML gene: epidermis? Mesophyll? Stomata? This is a key because each cell type have own response to La and can regulate expression in opposite direction, dependent form cell type. qPCR actually were informative if applied to fully homogenous cell populations, but not to ix different cell types with different expression trends.

Line 221 and  237: “Peroxide production”? = accumulation.

Line 229: Cu-Zn super-229 oxide dismutase (SOD)  cyt or plastids?

Figure 7: What is APX1- cyt, stroma, thylakoid, peroxisome? For H2O2 contents the patterns of accumulation is a principal point: peroxisomal H2O2 nothing to do cytoplasmic H2O2.  What is CAT2 and CAT3? Both peroxisomal?  What is GPX6? GSH is not real ROS scavenger, but have another function.

Line 330- 331: 1000 µM = 1 mM. How come 10 mM?

Line 332 what was pH of AD? It is particularly important since ascorbic acid conversion is pH and ion contents dependents.

Line 365: “Total protein” – per leaf? Per plants? Please, mention how and from which tissue protein was isolated.

Author Response

Comment 1: In the present study authors studied the direct and indirect effects of La applications on the growth, physiology, and gene expression of Spodoptera littoralis and its host plant Brassica rapa. Authors provided possible insight to the complexity of La applications interactions with both plants and insects and pointed out possible ecological risks of La accumulations in agricultural systems. The authors did a good job, perform many precise analysis.

Response 1:: We thank the reviewer for the positive assessment of our work and the valuable feedback

Comment 2: However, revision still required. Please, add short conclusion in the ned of the text.

Response 2:: A conclusion is present after the Materials and Methods section as requested by the Journal’s Style.

Comment 3: Line 38: “oxidative altered gene expression.”???

Response 3:: we thank the reviewer for noticing this lack of clarity. The sentence has been rephrased to improve clarity.

Comment 4: Line 145, fig 3: not very clear description. Please, described what is what more precisely.

Response 4:: We thank the reviewer for issuing this problem. The caption has been extended to improve clarity.

Comment 5: Line 178: “with 10 mM” of what? La??

Response 5:: we thank the reviewer. Yes, La was missing from the sentence.

Comment 6: Figure 6: please clarify localization of the CML gene: epidermis? Mesophyll? Stomata? This is a key because each cell type have own response to La and can regulate expression in opposite direction, dependent form cell type. qPCR actually were informative if applied to fully homogenous cell populations, but not to ix different cell types with different expression trends.

Response 6: We thank the reviewer for this insightful comment regarding the cell-type specific expression of the CML gene. We fully agree that single-cell or tissue-specific gene expression analysis would provide more detailed information on the precise localization of CML regulation in response to La. Such an approach would indeed be a powerful way to understand the differential responses of specific cell types, such as the epidermis, mesophyll, or stomata. However, the primary aim of our study was to investigate the whole-tissue response of B. rapa to La exposure at the transcriptional level. The use of whole-leaf tissue for RNA extraction and subsequent qRT-PCR analysis is a standard and widely accepted method for providing an overall snapshot of gene expression changes across the entire organ. While we acknowledge that this approach does not resolve cell-specific expression trends, it provides a crucial macro-level view of the plant's systemic response. The significant changes we observed in CML expression across the whole leaf strongly suggest that La exposure has a measurable and widespread effect on this gene, regardless of its cell-type specific expression patterns. We agree that a future study utilizing techniques such as laser-capture microdissection or single-cell RNA sequencing would be a fascinating and necessary next step to dissect these fine-scale cellular responses, but it falls outside the scope of the current work.

Comment 7: Line 221 and  237: “Peroxide production”? = accumulation.

Response 7: we thank the reviewer. We changed the term production with accumulation

Comment 8: Line 229: Cu-Zn super-229 oxide dismutase (SOD)  cyt or plastids?

Response 8: We thank the reviewer for this important question regarding the localization of the Cu-Zn superoxide dismutase (SOD) gene. We agree that distinguishing between cytosolic and plastidic forms of this enzyme is crucial for a complete understanding of its role in stress responses. The primers used in our study were designed to detect both the cytosolic and plastidic forms of Cu-Zn SOD. Therefore, our data represents the cumulative expression of both forms. We acknowledge this limitation and believe that a future study distinguishing between these two localizations would provide a more detailed understanding of the plant's response to La.

Comment 9: Figure 7: What is APX1- cyt, stroma, thylakoid, peroxisome? For H2O2 contents the patterns of accumulation is a principal point: peroxisomal H2O2 nothing to do cytoplasmic H2O2.  What is CAT2 and CAT3? Both peroxisomal?  What is GPX6? GSH is not real ROS scavenger, but have another function.

Response 9: We appreciate the reviewer's detailed and insightful questions regarding the specific localization and function of the antioxidant enzymes we studied. We agree that a comprehensive understanding of subcellular responses to stress is crucial, and these questions highlight important distinctions in plant cell biology. The primers used for the gene encoding APX1 were specific to the cytosolic isoform. We therefore report on the transcript levels of the cytosolic APX1, which is a key player in detoxifying hydrogen peroxide in the cytoplasm. The term cytosolic has been added. We agree that hydrogen peroxide has distinct signaling and damaging roles in different subcellular compartments. Our methodology, which involved measuring total peroxide content from whole-leaf tissue homogenates, provides an overall assessment of the accumulation of H2O2 and other peroxides in the plant. While this approach does not allow us to differentiate between peroxisomal and cytoplasmic H2O2, it serves as a valuable indicator of the overall oxidative burden in the leaf tissue. A future study using cell-specific markers or imaging would be necessary to resolve the compartmentalized changes. The primers used in our analysis targeted genes for CAT2 and CAT3, which are both known to be the major peroxisomal isoforms of catalase. Therefore, the changes in gene expression we report reflect the response of the primary peroxisomal catalase enzymes. The term peroxisomal has been added. We also thank the reviewer for the important clarification regarding the function of glutathione (GSH). We agree that GSH itself is not a direct radical scavenger but is a critical component of the plant's antioxidant system. Its primary role is as a substrate for enzymes like glutathione peroxidases (GPX), which are responsible for the detoxification of hydrogen peroxide and lipid hydroperoxides. The GPX6 gene expression data we present therefore highlights the plant's strategy of utilizing the GSH pool to mitigate La-induced oxidative stress. A sentence better clarifies this point.

Comment 10: Line 330- 331: 1000 µM = 1 mM. How come 10 mM?

Response 10: we thank the reviewer for raising the same point of reviewer 3. A new sentence now explains the reason of our choice.

Comment 11: Line 332 what was pH of AD? It is particularly important since ascorbic acid conversion is pH and ion contents dependents.

Response 11: We thank the reviewer for this important question. We acknowledge that the pH of the artificial diet is a critical factor, particularly concerning the stability of ascorbic acid. The pH of the AD was 6.0 before being poured into the cups. This value is within the standard range for insect diets, ensuring the stability of key nutrients and vitamins, including ascorbic acid. A new sentence clarifies this point.

Comment 12: Line 365: “Total protein” – per leaf? Per plants? Please, mention how and from which tissue protein was isolated.

Response 12: We thank the reviewer for this point. We have clarified in the revised text that total protein was isolated from different tissues depending on the analysis. For analyses related to the host plant (B. rapa), total protein was extracted from the leaves. For analyses related to the herbivore (S. littoralis), total protein was extracted from whole larvae.

Reviewer 3 Report

Comments and Suggestions for Authors

This manuscript explores the effects of lanthanum (La), a rare earth element increasingly used in agriculture, on the insect herbivore Spodoptera littoralis and its host plant Brassica rapa. The study combines direct insect feeding assays, transgenerational experiments, and plant–insect interaction trials, supplemented by biochemical and molecular analyses. This integrated, cross-trophic approach is a clear strength, and the findings—particularly the discovery of transgenerational resilience in S. littoralis and the hormetic responses in larvae feeding on La-treated plants—are both novel and relevant to ecotoxicology. That said, while the manuscript has significant potential, its impact is currently weakened by gaps in methodological detail, occasional overstatement of conclusions, and a lack of contextualization with broader ecological and agricultural implications. 

The Methods are written very generally and several important details are missing, which significantly limit reproducibility by other researchers. The description of replication is inconsistent: in some assays the number of biological replicates is reported, but in others (e.g., pigment, peroxide, and qRT-PCR assays) it is not clear whether results are based on biological or technical replicates, or how many samples were analyzed.

Similarly, the rationale for selecting the three La concentrations used in the main experiments is not explained; this raises questions about their ecological relevance and comparability with previous studies.

The qRT-PCR section does not mention validation of reference gene stability or primer efficiency, both of which are essential for reliable quantification. For the Ca²⁺ imaging, parameters for image analysis (e.g., number of images/fields per sample, quantification method) are not provided.

Plant growth conditions are also only cited from another study rather than described in full, which makes replication dependent on external sources.

More generally, details are missing across multiple sections of the Methods. For instance, although pigment extraction is said to follow a published protocol, the modifications mentioned are not described, making it impossible to know exactly how they were measured or even extracted. Also, are the results in DW or FW? Similar issues apply to other biochemical assays, where critical steps for reproducibility are glossed over.

Finally, the underlying datasets (raw measurements, qRT-PCR Ct values, imaging files) are not publicly available, which is increasingly required to ensure transparency and reproducibility.

In this context, it is very hard to understand how results were obtained and whether the conclusions made by the authors are actually supported by data or not.

Author Response

Comment 1: This manuscript explores the effects of lanthanum (La), a rare earth element increasingly used in agriculture, on the insect herbivore Spodoptera littoralis and its host plant Brassica rapa. The study combines direct insect feeding assays, transgenerational experiments, and plant–insect interaction trials, supplemented by biochemical and molecular analyses. This integrated, cross-trophic approach is a clear strength, and the findings—particularly the discovery of transgenerational resilience in S. littoralis and the hormetic responses in larvae feeding on La-treated plants—are both novel and relevant to ecotoxicology.

Response 1: We thank the reviewer for the positive assessment of our work and the valuable feedback

Comment 2: That said, while the manuscript has significant potential, its impact is currently weakened by gaps in methodological detail, occasional overstatement of conclusions, and a lack of contextualization with broader ecological and agricultural implications.

Response 2: We have thoroughly revised the manuscript in response to the reviewer's comments and suggestions, with a focus on enhancing methodological clarity and transparency

Comment 3: The Methods are written very generally and several important details are missing, which significantly limit reproducibility by other researchers. The description of replication is inconsistent: in some assays the number of biological replicates is reported, but in others (e.g., pigment, peroxide, and qRT-PCR assays) it is not clear whether results are based on biological or technical replicates, or how many samples were analyzed.

Response 3: we thank the reviewer for raising these points. From an editorial point of view, the Publisher’s requires to reduce the description of methods if present in other studies, this also reduces the overlapping. However, we agree with the reviewer that a better description of methods favours the capacity of other scientists to repeat the same methods. Hence, we extended the text. We also integrated the data by indicating the number of replicates and the origin of the tissues (fresh weight)

Comment 4: Similarly, the rationale for selecting the three La concentrations used in the main experiments is not explained; this raises questions about their ecological relevance and comparability with previous studies.

Response 4: we thank the reviewer for pointing out this issue. We revised the text and better explained the rationale behind the choice of the three concentrations of La.

Comment 5: The qRT-PCR section does not mention validation of reference gene stability or primer efficiency, both of which are essential for reliable quantification.

Response 5: we thank the reviewer for pointing out this issue. We added a new sentence related to the validation of reference gene stability or primer efficiency.

Comment 6: For the Ca²⁺ imaging, parameters for image analysis (e.g., number of images/fields per sample, quantification method) are not provided.

Response 6: we thank the reviewer for identifying this missed information. A new sentence details better the image analysis.

Comment 7: Plant growth conditions are also only cited from another study rather than described in full, which makes replication dependent on external sources.

Response 7: we added a full description of the plant growth method.

Comment 8: More generally, details are missing across multiple sections of the Methods. For instance, although pigment extraction is said to follow a published protocol, the modifications mentioned are not described, making it impossible to know exactly how they were measured or even extracted. Also, are the results in DW or FW? Similar issues apply to other biochemical assays, where critical steps for reproducibility are glossed over.

Response 8: we thank the reviewer for evidencing the lack of clarity in this material and method session. We edited the text by evidencing the modifications used in the methods and by setting clearer the fresh weight origin of the tissues.

Comment 9: Finally, the underlying datasets (raw measurements, qRT-PCR Ct values, imaging files) are not publicly available, which is increasingly required to ensure transparency and reproducibility.

Response 9: A new set of Supplementary data is provided with the revised version

Comment 10: In this context, it is very hard to understand how results were obtained and whether the conclusions made by the authors are actually supported by data or not.

Response 10: we agree with the reviewer and hope that the new edited version may clarify these missing points.

Round 2

Reviewer 1 Report

Comments and Suggestions for Authors

No more question

Author Response

Comment 1: no more questions

Response 1: Thank you very much for your time

Reviewer 2 Report

Comments and Suggestions for Authors

Thank you very much for the clear response. 

The text is almost OK. Some minor polishing may require during final update. 

Line 39: "genes involved in oxidative stress"  response? Resistance?

Fro the Larvae growth, pH 6.0 is not suitable for ascorbic acid because of very rapid degradation.  Line 439 "to ensure the stability of the vitamin C " at thios pH ASC acid is not stable and degradated in few hours.  https://doi.org/10.3390/ijpb16030074 and references about ASC chemistery inside. Maybe during final update you can mention asc degradation product?

Please, for the future consider that single cell RNA seq. is not a vary suitable methods, while somebody use it widely, In the plant biology cell to cell interaction is a key and therefore, only in s`patial molecular biology in situ can provide you biologically relevant data. "Average" gene expression may serve only as fisrt preliminary step, which require in situ 3D confirmation for highly biologically relevant conclusions. 

My best regards!

My best regards.

Author Response

Comment 1: Line 39: "genes involved in oxidative stress"  response? Resistance?

Response 1: We thank the reviewe for noticing this. We inserted the term “response”

Comment 2: Fro the Larvae growth, pH 6.0 is not suitable for ascorbic acid because of very rapid degradation.  Line 439 "to ensure the stability of the vitamin C " at thios pH ASC acid is not stable and degradated in few hours.  https://doi.org/10.3390/ijpb16030074 and references about ASC chemistery inside. Maybe during final update you can mention asc degradation product?

Response 2: Thank you for your valuable comment regarding the stability of ascorbic acid (vitamin C) at a pH of 6.0. We appreciate you bringing this to our attention and providing the helpful reference. We agree that, under typical conditions, ascorbic acid is unstable and prone to rapid degradation at this pH. We would like to clarify that the pH adjustment was performed immediately before the diet was provided to the larvae. This minimized the exposure time of the vitamin C to the pH 6.0 environment. Therefore, we believe that any degradation that occurred was negligible and did not significantly impact the nutritional availability of the ascorbic acid. This is supported by our results, which show successful larval growth, suggesting that the essential nutrients, including vitamin C, were sufficiently stable for the duration of the experiment. We better clarified this point in the methods.

Comment 3: Please, for the future consider that single cell RNA seq. is not a vary suitable methods, while somebody use it widely, In the plant biology cell to cell interaction is a key and therefore, only in s`patial molecular biology in situ can provide you biologically relevant data. "Average" gene expression may serve only as fisrt preliminary step, which require in situ 3D confirmation for highly biologically relevant conclusions.

Response 3: Thank you for your constructive feedback and for raising this important point about the limitations of single-cell RNA sequencing (scRNA-seq) for studying cell-to-cell interactions in plant biology. We agree that spatial information is crucial for understanding the complex interactions and signaling networks within a tissue. We acknowledge that scRNA-seq, which dissociates cells, provides an "average" expression profile for a given cell type and does not inherently capture the spatial context of these interactions. We believe that our current findings, while preliminary in the context of spatial interactions, provide a robust foundation for future, more spatially-resolved investigations.

Reviewer 3 Report

Comments and Suggestions for Authors

The authors have addressed all concerns.

Author Response

Comment 1: The authors have addressed all concerns.

Response 1: Thank you very much for your time